# Planktonic microbial signatures of sinking particle export in the open ocean's interior

Fuyan Li[1,2], Andrew Burger[1,2], John M. Eppley [1,2], Kirsten E. Poff[1,2,3], David M. Karl [1,2] & Edward F. DeLong [1,2] ✉

A considerable amount of particulate carbon produced by oceanic photosynthesis is exported to the deep-sea by the "gravitational pump" (~6.8 to 7.7 Pg C/year), sequestering it from the atmosphere for centuries. How particulate organic carbon (POC) is transformed during export to the deep sea however is not well understood. Here, we report that dominant suspended prokaryotes also found in sinking particles serve as informative tracers of particle export processes. In a three-year time series from oceanographic campaigns in the Pacific Ocean, upper water column relative abundances of suspended prokaryotes entrained in sinking particles decreased exponentially from depths of 75 to 250 m, conforming to known depth-attenuation patterns of carbon, energy, and mass fluxes in the epipelagic zone. Below ~250 m however, the relative abundance of suspended prokaryotes entrained in sinking particles increased with depth. These results indicate that microbial entrainment, colonization, and sinking particle formation are elevated at mesopelagic and bathypelagic depths. Comparison of suspended and sinking particle-associated microbes provides information about the depth-variability of POC export and biotic processes, that is not evident from biogeochemical data alone.

Solar energy absorbed by phytoplankton is converted into chemical energy and used to produce organic matter from carbon dioxide $(CO_2)$[1,2]. Globally, a considerable fraction (~18–20%)[3] of this net primary productivity is transported to the deep sea via sinking particles where it nourishes the deep-sea ecosystem and is sequestered from the upper ocean for centuries[3]. The export of particulate organic carbon (POC) to deep waters sequesters carbon in the deep sea and helps regulate the partitioning of $CO_2$ between the ocean and the atmosphere.

Physical, chemical, and biological processes all contribute to the vertical export of sinking POC to deeper waters[4–7]. In the upper water column, the POC flux decreases exponentially with depth. One commonly used empirical approach to describe particle flux attenuation with depth is known as the "Martin curve"[8]. The model is expressed as $F_Z = F_{100}(Z/100)^{-b}$, where $Z$ is the POC collection depth, $F_{100}$ is the reference POC flux at 100 m, and $b$ is a unitless parameter that represents the magnitude of flux attenuation with depth. The Martin curve

power-law relationship is typically used to describe POC flux attenuation between depths of ~100–2000 m[8], where sinking particle degradation typically exceeds its formation. While POC flux attenuation in the upper water column is well described, biological details relevant to particle production and consumption throughout the ocean's interior are less well understood.

Picophytoplankton, such as *Prochlorococcus* and *Synechococcus*, have been previously reported to contribute to POC export in the open ocean[9,10] in part via sinking phytodetrital macroaggregates and fecal pellets via the biological pump[11,12], and the "migrant pump"[3,5]. The seasonal dynamics of sinking particle-associated microbial genes and genomes reaching the abyss (including genes encoding oxygenic and aerobic anoxygenic photosynthesis, cyanobacteria nitrogen fixation, and proteorhodopsin-based photoheterotrophy)[13], has further demonstrated the temporal variability of surface-derived particle flux via the biological pump[13]. Cumulatively, these data suggest that the

[1]Department of Oceanography, University of Hawai'i at Mānoa, Honolulu, HI, USA. [2]Daniel K. Inouye Center for Microbial Oceanography: Research and Education, University of Hawai'i at Mānoa, Honolulu, HI, USA. [3]Present address: McDonogh School, Owings Mills, MD, USA. ✉e-mail: edelong@hawaii.edu

genomic signatures of prokaryotes entrained within sinking particles may record depth-variable export processes throughout the water column.

Recent studies of suspended microbial biomarkers have emphasized the vertical connectivity of taxa from surface waters to the abyss[14,15], and an apparent increase in particle-attached microbial adaptations associated with suspended particles sampled at greater depths[16]. The relationships of dominant microbial taxa found in suspended microbial assemblages, to those taxa found on sinking particles, however, are less well constrained. To better understand particle export processes in the ocean interior, we compared 16S rRNA gene amplicon sequence variants (ASVs) of suspended prokaryotes found throughout the water column (0–4000 m), with sinking particle-associated ASVs collected in sediment traps deployed at a variety of depths (150–500 m, or 4000 m).

Here, we aim to determine whether suspended prokaryotes once entrained in sinking particles can provide information about depth-specific particle export processes throughout the water column. Bacterial and archaeal ASVs (see Table 1) suspended in the water column were compared to those ASVs associated with sinking particles collected in sediment traps, at multiple depths (see Methods section). Suspended ASVs that were shared with (and identical to) sediment trap-collected ASVs are referred to herein as "Shared ASVs" (SASVs, Table 1). Identical SASVs that were present in both suspended as well as sinking-particle prokaryote assemblages collected in sediment traps, were then used to explore the vertical and temporal dynamics of sinking particle export (Supplementary Fig. 1). In the euphotic zone, the water column relative abundances of SASVs decreased with increasing depth, in accordance with known particle flux attenuation at these depths. In mesopelagic and bathypelagic zones, both water column and sediment trap relative abundances of SASVs increased with depth, suggesting increased prokaryote colonization and growth on deep-sea sinking particles, and elevated new sinking particle formation in the deep ocean.

## Results and discussion
### Attenuation curves of upper water column SASV relative abundances

A total of 3902 SASVs were identified from comparisons of all 4000 m sediment trap ASVs (22,803 total ASVs), with all the suspended water column ASVs (54,990 total ASVs) (Supplementary Fig. 1). Similarly, for the PARAGON summer cruise samples (see Methods section), analyses of 5586 water column ASVs and 2186 500 m sediment trap-collected ASVs, resulted in identification of 1487 SASVs (Supplementary Fig. 1).

The water column relative abundances of the 4000 m trap SASVs (Table 1) were highest in shallower waters, ranging between 35.6% to 70.3% at and above 75 m with mean values of 56.5–57.9% above 75 m and 53.3% at 75 m (Fig. 1a, d, Supplementary Data 1). These relative abundances dramatically decreased to 15.3–67.4% with mean values of 46.5% and 31.4% at 100 m and 125 m, respectively (Fig. 1a, d, Supplementary Data 1). Between 150 to 225 m, the 4000 m trap SASVs

decreased slightly to 14.2–45%, with mean values of 27.1% at 200 m to 28.6% at 150 m (Fig. 1a, d, Supplementary Data 1).

The upper water column relative abundances of the 500 m trap SASVs exhibited similar patterns of depth-dependent attenuation with increasing depth (Fig. 1b, c, Supplementary Data 1). The 500 m trap SASVs ranged from 80.3–92.3% with mean values of 82.9% (75 m) to 89.9% (25 m) at and above 75 m, and below 75 m decreased to 15.9–32.9% with mean values of 18.8% (300 m) to 30.2% (150 m).

The average water column abundances of SASVs ($S_Z$) at a given depth ($Z$(m)) for both 4000 trap and 500 m trap SASVs, displayed a depth-dependent decrease with optimal fits to a power-law function of the form $S_Z = S_{75} (Z/75)^b$ ($R^2$ values were 0.90 and 0.96, respectively for the 4000 m and 500 m trap SASVs; Fig. 1c, d). These depth trends are similar to those seen in the "Martin curve"[8] at Station ALOHA. For our SASV data, the best power-law fits were obtained using 75 m as the reference depth (see Methods). Sediment traps deployed at other depths (175, 200, 250 and 300 m) also showed similar SASV abundance trends, compared to the 4000 m deep traps, with similarly good fits to the same power-law function. (Supplementary Fig. 2, Supplementary Data 1).

Can quantitative data on suspended microbial entrainment in sinking particles provide useful information on the dynamics and biology of POC export? Although the methodologies we used were quite different from prior studies, the SASV attenuation curves recapitulated depth-dependent trends that have been previously reported for chlorophyll, POC, energy (enthalpy, or specific energy content of particles in Joules per unit mass), and mass depth attenuation in the NPSG[1,4] (Fig. 2). Notably, the biogeochemical data we used here from Grabowski et al.[1] (Fig. 2), were collected six months prior to the 4000 m trap SASV time-series data we report here from the same study site. The trends observed were consistent for time-averaged data collected over three years (in sediment traps deployed at 4000 m), as well as those deployed for shorter time periods at a variety of shallower depths (175, 200, 250, 300 and 500 m) (Fig. 2, Supplementary Fig. 2). A notable difference between the deep and shallow traps however, is that the attenuation of SASVs in the upper water column appear slightly more pronounced in shallow traps compared to deep traps (Fig. 2, Supplementary Fig. 2). The lesser contribution of upper water column ASVs to the 4000 m deep trap SASVs, (Fig. 1c–e), likely reflects the elevated degradation of particles that originate from surface waters[1,7]. These analyses are consistent with the known trend of sinking particle consumption exceeding its production at these depths at Station ALOHA (75-250 m)[1,7]. These results in total support the proposal that analyses of suspended microbes that are entrained in sinking particles at different depths, can yield useful information that is relevant to sinking particle degradation, formation and biotic processes.

SASV attenuation curves from the 4000 m sediment traps showed greater variation in b coefficients (from −1.26 to −0.44 over time) than did shallower sediment traps (Supplementary Fig. 3a and

## Table 1 | Definitions of terminology

| | |
|---|---|
| ASVs | 16S ribosomal RNA gene Amplicon Sequence Variants, measured in either suspended particulate matter in the water column, or in sinking particles collected in sediment traps. |
| SASVs | Shared ASVs that are found in both suspended particulate matter in the water column, and in sinking particles captured in sediment traps. |
| 500 m sediment trap SASVs | Shared ASVs found in both suspended particulate matter and sinking particles collected in 500 m sediment traps. |
| 4000 m sediment trap SASVs | Shared ASVs found in both suspended particulate matter and sinking particles collected in 4000 m sediment traps. |
| Water column relative abundance of SASVs | The relative abundance of sediment trap-shared SASVs in the water column at specific depths. |
| Sediment trap relative abundance of SASVs | The relative abundance of sediment trap-shared SASVs in the sediment traps deployed at specific depths. |

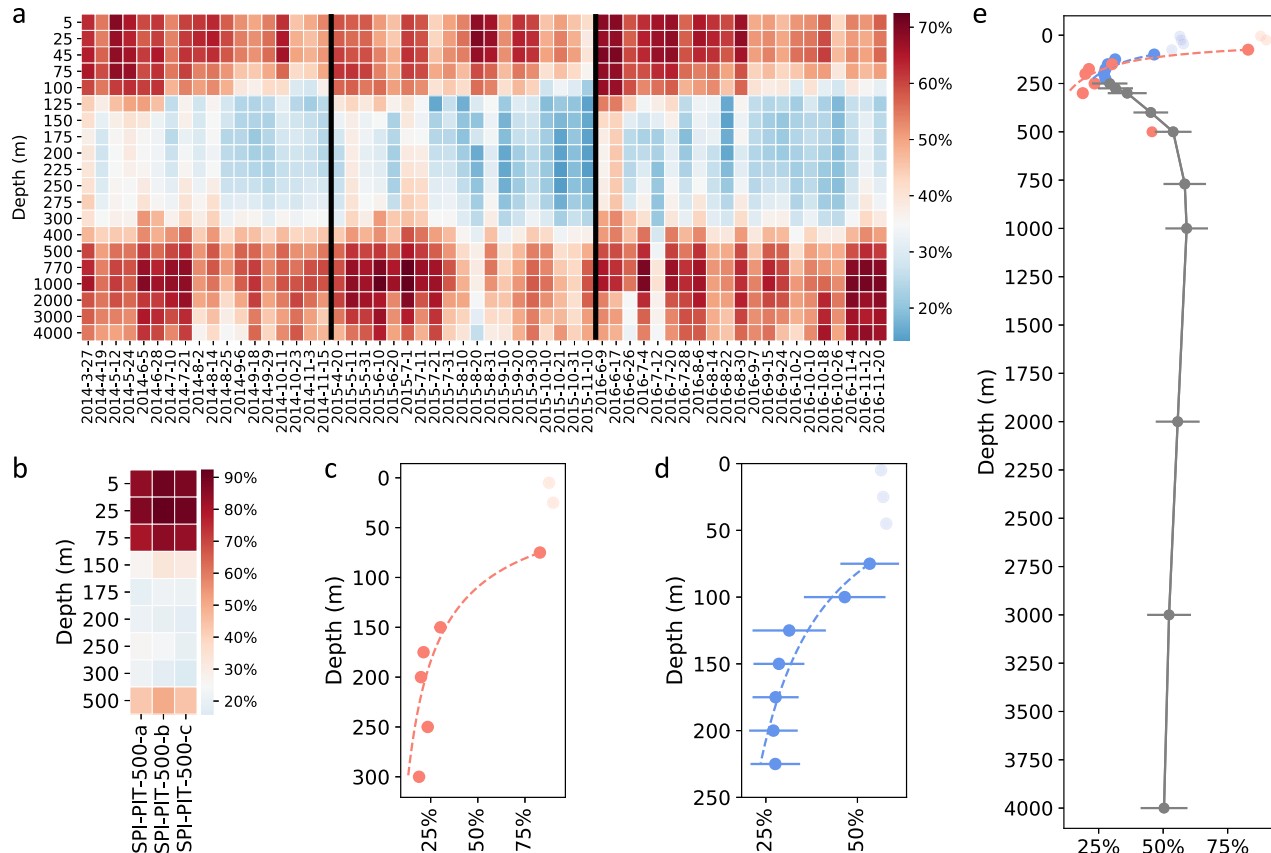

**Fig. 1 | Depth and temporal profiling of SASVs expressed as a function of their relative abundances at different depths in the water column. a** Depth and temporal profiles of 4000 m trap SASV relative abundances in the water column during the entire 4000 m trap deployment time period. (see Methods for details). **b** Depth profiles of PARAGON 500 m trap SASV relative abundances in the water column using three biological replicates, **a–c**. **c** Depth profiles of PARAGON 500 m trap SASV average relative abundances in the upper water column (≤300 m). The dashed red curve represents the best fit to the time-averaged data, and was calculated as: $S_Z = S_{75} (Z/75)^b$, where $R^2 = 0.96$, $S_{75} = 82.9$ (%), $b = -1.34$ (excluding light red symbols (<75 m)). The horizontal red bars represent the standard deviations ($n = 3$, Supplementary Data 1), and since they are smaller than the data point symbols in **c**, are not visible. **d** Depth profiles of 4000 m trap SASV average relative abundances in the upper water column (<250 m). The dashed blue curve represents the best fit to the time-averaged relative abundances, calculated as: $S_Z = S_{75} (Z/75)^b$, where $R^2 = 0.90$, $S_{75} = 53.11$(%), $b = -0.75$ (excluding light blue symbols (<75 m)). The horizontal blue bars represent the standard deviations ($n = 58$, Supplementary Data 1). **e** Depth profile of 4000 m trap time-averaged SASV relative abundances in the water column. The dashed red and blue curves correspond to those shown in **c** and **d**, respectively. The gray symbols represent 4000 m trap SASVs found in deeper waters (≥250 m) and the horizontal gray bars represent the standard deviations ($n = 58$, Supplementary Data 1).

Supplementary Data 2). Power-law curves of 4000 m trap SASVs from all time intervals also exhibited significant decay ($p$ value < 0.05; 95% confidence intervals are shown in Supplementary Fig. 3b and Supplementary Data 2). These features were similarly observed in PARAGON shallow trap deployments (Supplementary Fig. 4). The b coefficients of Station ALOHA time-series SASV power-law curves likely reflect the change in organic matter remineralization rates[1,7]. Dissolved inorganic phosphate, nitrite+nitrate and silicate exhibited the greatest negative correlations with b coefficients within and below DCM (Supplementary Fig. 3c), probably a reflection of the greater remineralization of particulate organic matter and thus the enhanced concentration of dissolved inorganic nutrients in the upper water column[17]. Significant correlations between temperature and Martin curve b coefficients were not observed, likely because most particle degradation in the NPSG occurs in the warm upper water column.

### Increases in mesopelagic and bathypelagic SASV water column relative abundances

Below 250 m, SASV water column abundance depth trends were notably different. Specifically, 4000 m trap SASV water column abundances were lowest at 250–300 m, varying between 14.6–53.1%, with mean values of 29.3% (250 m) to 36.1% (300 m). The 4000 m trap

SASV representation increased for suspended microbes originating from 400 and 500 m, ranging between 28.9–64.7%, with a mean of 45.2% at 400 m and 53.9% at 500 m. The 4000 m trap SASV representation increased slightly to 38.3–72.4% at greater depths, with a mean of 58.4% at 770 m and 59.1% at 1000 m (Fig. 1a, e, Supplementary Data 1). Below 1000 m, 4000 m trap SASV water column abundances decreased slightly, to 26.2–69.2% with mean values of 50.3% (4000 m) to 55.6% (2000 m) (Fig. 1a, e; Supplementary Data 1). Notably, the water column maximum of SASVs between 770–1000 m was adjacent to the previously reported depth maximum of the particulate adenosine triphosphate (ATP)[18,19], consistent with elevated sinking particle production and microbial activities that have been postulated for this depth horizon[20,21].

### Prokaryote community composition of 4000 m trap SASVs in the water column

The most abundant 4000 m trap SASVs in the water column (those ≥0.1% at any given depth), clustered together into discrete depth zone categories (Fig. 3a, Supplementary Data 3; see methods), as follows: Surface SASVs with maximum water column abundances between either 5–75 m (Surface), 100–150 m (DCM), 175–200 m (Lower Euphotic), 225–500 m (Upper Mesopelagic), 770–1000 m (Lower

Mesopelagic) and 2000–4000 m (Bathypelagic; Fig. 3a, Supplementary Data 3). Collectively, these individual SASVs represented 71.0–94.0% of their total water column abundances (Fig. 3b).

As expected, SASVs from surface waters were well represented by abundant photoautotrophic cyanobacteria like *Prochlorococcus* and

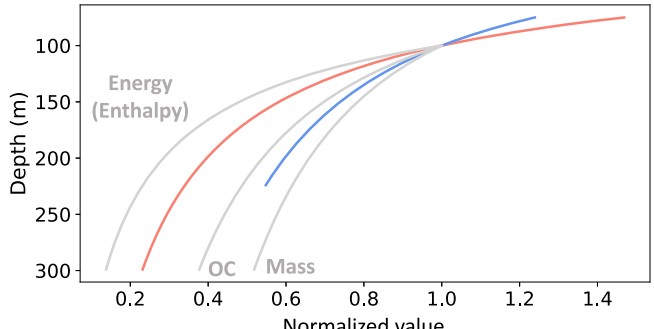

**Fig. 2 | Schematic representation of power-law curves determined for 4000 m trap (blue curve) and PARAGON 500 m trap (red curve) SASVs in the upper water column.** These are compared with previously published normalized attenuation flux data of energy, organic carbon and mass (gray curves), as reported in Grabowski et al.[5].

*Synechococcus*, reflecting their passive entrainment with sinking particles that reach greater depths[22]. Prokaryotic SASVs from 175 to 200 m included oligotrophic heterotrophs like SAR324, SAR11, and chemolithoautotrophic archaeal nitrifiers like *Nitrosopelagicus*. Between 225–500 m, similar trends were observed with SAR324, SAR406, and Nitrosopumilaceae SASVs predominating. Likewise, in the water column below 500 m, SAR324, SAR406 and Nitrosopumilaceae SASVs were the most abundant SASVs (Fig. 3b). The notable increase in deeper water SASVs that are affiliated with chemolithotrophic taxa (mainly nitrifiers and sulfur oxidizers), is consistent with known trends that have been observed in previous studies[20,21,23–26]. It has been proposed that such chemolithotrophs may participate in the production of new particulate organic carbon especially at depths of 700–900 m[20,21,23], and therefore are involved deep-sea particle export dynamics at this depth horizon. Collectively, these data show that regardless of depth of origin, the most abundant suspended prokaryote ASVs in the water column were the same ASVs found within sinking particles reaching 4000 m (Supplementary Fig. 5). Mechanistically, these data are proposed to reflect several processes that include: 1) The passive entrainment of deeper water suspended prokaryotes into sinking particles formed in the upper water column; and 2) The incorporation of deep-sea prokaryotes into sinking particles that are formed in situ via active colonization or passive entrainment (Supplementary Fig. 6).

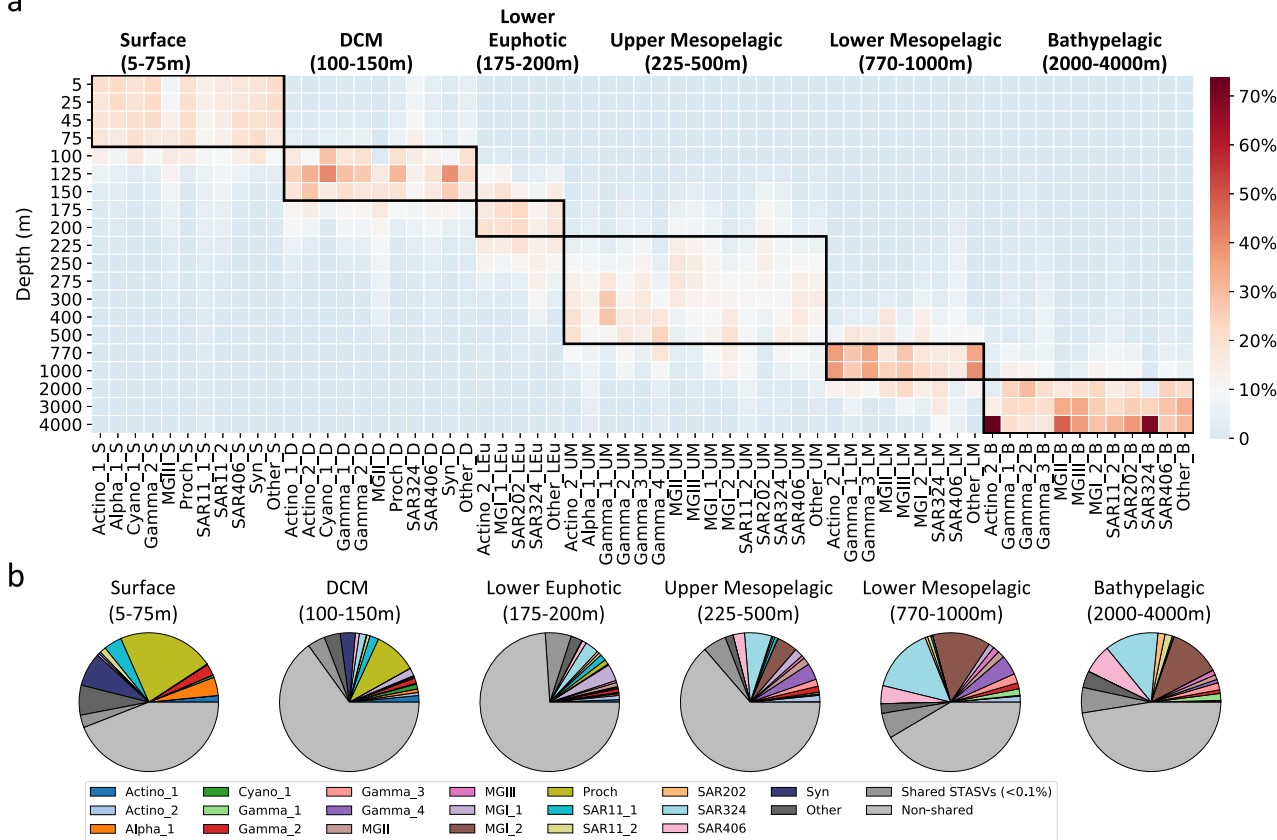

**Fig. 3 | Taxonomic profiles of 4000 m trap SASV relative abundances in the water column. a** Depth profiles of 4000 m trap SASV taxa at specific water column depths, relative to their time-averaged abundances throughout the water column (see methods). **b** Community composition of 4000 m trap SASVs in specific depth strata. Actino_1: Actinobacteriota (Candidatus Actinomarina); Actino_2: Actinobacteriota (Sva0996 marine group); Alpha_1: Alphaproteobacteria (AEGEAN-169 marine group); Cyano_1: Cyanobacteria (Unknown Cyanobiaceae); Gamma_1: Gammaproteobacteria (HOC36); Gamma_2: Gammaproteobacteria (SAR86 clade); Gamma_3: Gammaproteobacteria (UBA10353 marine group); Gamma_4:

Gammaproteobacteria (Unknown); MGII: Marine Group II Poseidoniia; MGIII: Marine Group III Poseidoniia; MGI_1: Marine Group I Thaumarchaeota (Candidatus Nitrosopelagicus); MGI_2: Marine Group I Thaumarchaeota (Unknown Nitrosopumilaceae); Proch: Prochlorococcus; SAR11_1: SAR11 (Clade Ia); SAR11_2: SAR11 (Clade II); SAR202: Chloroflexi (SAR202 clade); SAR324: SAR324 clade (Marine group B); SAR406: Marinimicrobia (SAR406 clade); Syn: Synechococcus. Depth-specific ecotypes: _S, Surface; _D, DCM; _LEu, Lower Euphotic; _UM, Upper Mesopelagic; _LM, Lower Mesopelagic; _B, Bathypelagic.

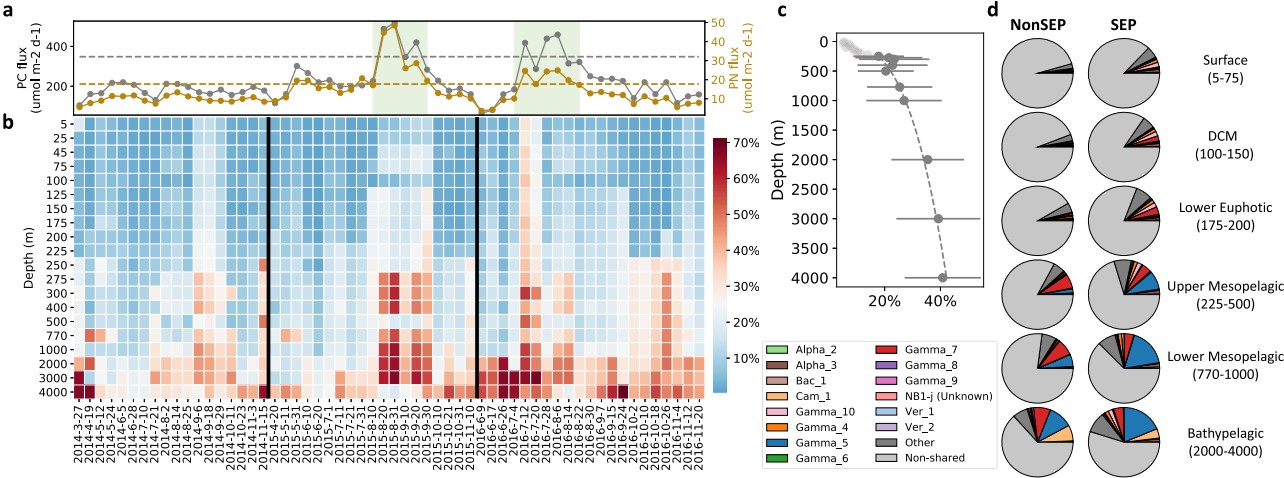

**Fig. 4 | Depth and temporal profiles of SASVs found at discrete depths in the water column, expressed as a function of their relative abundances in the 4000 m deep traps. a** Previously reported particulate carbon (gray symbols) and particulate nitrogen (brown symbols) fluxes in the 4000 m deep traps[13]. The dashed horizontal gray and brown lines are equal to 150% of the respective annual mean values observed between 1992 and 2004 at Station ALOHA[7]. The green-shaded time-period corresponds to the SEP. **b** Time-averaged 4000 m trap SASV relative abundances in the trap, mapped to the depths where corresponding identical water column SASVs are found. (see Methods for details). **c** Data from **b**, representing the best fit to power-law function of the form $S_Z = S_{250} (Z/250)^b$ ($R^2 = 0.93$, $S_{250} = 17.76$ (%), $b = 0.31$), excluding light gray symbols at shallower depths (<250 m). $S_Z$ is time-averaged relative abundance of 4000 m trap SASVs as a function of their relative abundances in the deep traps at a depth $Z$(m). $S_{250}$ is $S_Z$ at

250 m, and $b$ is the coefficient of change of deep trap relative abundances of SASVs at specific depths. The horizontal gray bars represent the standard deviations ($n = 58$, Supplementary Data 4). **d** Composition of prokaryotic SASVs in the 4000 m deep sediment traps during non-SEP and SEP time periods. Alpha_2: Alphaproteobacteria (Hyphomonas); Alpha_3: Alphaproteobacteria (Unknown Rhodobacteraceae); Bac_1: Bacteroidota (Unknown Saprospiraceae); Cam_1: Campilobacterota (Halarcobacter); Gamma_10: Gammaproteobacteria (Unknown Vibrionaceae); Gamma_4: Gammaproteobacteria (Unknown); Gamma_5: Gammaproteobacteria (Colwellia); Gamma_6: Gammaproteobacteria (Halioglobus); Gamma_7: Gammaproteobacteria (Moritella); Gamma_8: Gammaproteobacteria (Psychrobium); Gamma_9: Gammaproteobacteria (Shewanella); Ver_1: Verrucomicrobiota (Unknown DEV007); Ver_2: Verrucomicrobiota (Unknown Puniceicoccaceae).

## Increased deep water SASV contributions to sinking particles collected in the 4000 m traps

When the relative abundances of SASVs in the sediment traps themselves were considered (in contrast to the SASV water column proportions of Fig. 1), those originating from greater water column depths had greater proportions in the 4000 m traps (Fig. 4). SASVs from 5 to 225 m contributed the lowest proportion to the 4000 m traps, ranging from 0.6 to 37.8% with mean values of 5.2% at 25 m to 14.0% at 225 m (Fig. 4a, b; Supplementary Data 4). Those from 250 to 500 m varied between 2.1–59.8% with mean values of 17.9% at 250 m to 23.4% at 300 m, while those originating from 770 to 3000 m increased to 7.5–64.6% with mean values of 25.4% (770 m) and 39.2% (3000 m) (Fig. 4a, b; Supplementary Data 4). Those originating from 4000 m in the water column, contributed the most to 4000 m trap SASVs, varying from 18.1–71% with a mean of 40.8% (Fig. 4a, b; Supplementary Data 4). The observed increases in trap abundances contributed by SASVs at and below 250 m were fit to a power-law function of the form: $S_Z = S_{250} (Z/250)^b$ (Fig. 4b). The observed trends are consistent with prior studies indicating that deep-sea prokaryotes have a greater proportion of particle-associated adaptations[16] and support their proposed involvement in sinking particle colonization and growth at meso- and bathypelagic depths.

## Deep sea copiotroph contributions to sinking particles collected in the 4000 m traps

Specific SASVs of *Colwellia*, *Moritella* and *Halarcobacter* that originated from shallower waters had the lowest representation of these genera in the 4000 m traps (5–225 m, mean of 1.2%). The SASVs from the same genera originating from deeper waters increased in their trap representation (250 m, mean of 7.7%), with the highest trap-abundance values originating from 4000 m (mean of 28.8% in traps; Fig. 4c). Notably, *Colwellia* and *Moritella* contain genomic features and metabolic pathways that may enable colonization of sinking particles,

growth in anaerobic niches, and degradation of biopolymers[14,27]. In aggregate, the above data indicate substantial colonization and growth of copiotrophs on sinking particles at mesopelagic and bathypelagic depths[13,28].

## Seasonal change of station ALOHA time-series SASV trap relative abundances

To characterize seasonal patterns of sinking particles at Station ALOHA[7,18], we examined temporal patterns of SASVs as a function of their relative abundances in the 4000 m deep traps during the summer export pulse (SEP; when particulate nitrogen and carbon flux at 4000 m exceed annual mean fluxes by 150% or more[7]), versus non-SEP time periods (Fig. 4a, b). The SASV 4000 m trap abundances varied from 0.6 to 71.0% with mean values of 3.7–42.3% during non-SEP time period and were elevated to 1.6–68.3%, with mean values of 10.1–53.9% during the SEP (Fig. 4a, b, Supplementary Data 4). A total of 205 deep trap SASVs were enriched ($p < 0.05$) in the deep trap during the SEP compared to non-SEP time periods (Supplementary Data 5). Among these SASVs, the most abundant were affiliated with specific ecotypes of *Colwellia* and *Moritella* (Fig. 4d). Other deep trap-associated rarer taxa included Flavobacteriaceae, Rhodobacteraceae, Rhizobiales, *Erythrobacter*, Halieaceae, *Pseudoalteromonas*, Saprospiraceae, *Hyphomonas*, *Richelia* and *Crocosphaera* (Fig. 4d, Supplementary Data 5). These observations are consistent with previous findings based on metatranscriptomic gene transcript abundances previously reported in Poff et al.[13]. Collectively, the seasonal patterns observed in trap POC, PON, and particle-associated prokaryote taxa in the 4000 m sediment traps, reflect the annual variation of primary production and subsequent particle export flux.

In conclusion, our analyses show that in surface waters, the attenuation of oligotrophic surface-derived prokaryote taxa entrained within sinking particles in the upper water column, recapitulates known patterns of mass, chlorophyll, organic carbon, and energy

(enthalpy) associated with sinking particle flux[1,4]. At Station ALOHA, particle formation is driven by photosynthetic picoplankton like *Prochlorococcus* and *Synechococcus*, and larger eukaryotic phytoplankton[22,29,30]. The export of smaller picophytoplankton is facilitated by their packaging into fecal pellets that sink rapidly and may be more resistant to degradation[9–12,31,32]. Available data also suggest that picophytoplankton exported to deeper waters, is dominated by the most abundant genotypes from surface waters, since the most abundant surface-water *Prochlorococcus* ecotypes are also the most abundant types found at mesopelagic depths[33]. Taken together, our results suggest that the export of small phytoplankton to abyssal depths in the NPSG occurs year-round, but at elevated levels during the seasonal summer export pulse[7] (Supplementary Fig. 7).

In the mesopelagic and bathypelagic zones, the proportions of suspended prokaryotes entrained in sinking particles (including heterotrophic oligotrophs and copiotrophs, and chemolithotrophs) generally increased with increasing depth. The taxonomic richness of SASVs reaching 4000 m also increased, the greater the depth of SASV origin in the water column (Supplementary Fig. 8). These trends suggest increasing prokaryote entrainment, colonization and growth on sinking particles with depth, and increased new sinking particle formation in the deep-sea[20,21,23]. Some of these biological and ecological insights derived from such analyses cannot be derived from biogeochemical data alone. Mechanistically, protistan grazing[34], particle aggregation[5,11], and metazoan fecal pellet production[5,35], all contribute to deep sea sinking particle formation, and the consequent entrainment of suspended microbial taxa at greater depths. Sinking animal carcasses (e.g. fish, pteropods, jellies) from multiple depths are also known to contribute to sinking particle flux in the deep sea[13,28].

To constrain particle export and carbon sequestration processes in the open ocean, one proposed central research priority is to "better understand the mechanisms of particle transformations within a 4D framework"[5]. Our analyses confirm that suspended microbes entrained in sinking particles can provide depth-specific details about sinking particle origins and export dynamics. To further decipher the mechanisms of sinking particle formation and flux, improvements to this approach could include expanding spatiotemporal coverage using autonomous vehicles[36,37], diversifying regional analyses[38], obtaining more highly resolved taxonomic resolution for both suspended and sinking particle-attached prokaryotes[33] (including protist, metazoan and virus taxon dynamics[34,39]), and leveraging genome-resolved analyses[14] and other 'omics-enabled approaches[16,40]. Additional efforts might include expanding available datasets on the life histories and distributions of diverse microbes on individual sinking particles[41,42], and their specific associations with deep-sea protists and metazoan hosts[32,43,44]. Together, these and other strategies can provide a fuller understanding of the biological details and mechanisms that help regulate particle formation, degradation and colonization, which impact ecosystem dynamics and carbon export in the deep sea.

## Methods

### Sediment trap and water sample collections

Sinking particles reaching 4000 m were collected every 12 days on average, from a bottom-moored, sequencing sediment trap deployed at Hawaii Ocean Time-series (HOT) Station ALOHA. Samples were collected over a 3-yr time series, from Mar 2014 through Nov 2016, as previously described[13,28]. The sampling of sinking particles collected in the 4000 m trap time series, and subsequent particle recovery, processing, DNA extraction and 16S rRNA gene amplicon sequencing for sediment trap samples collected in 2014 have been previously described[28]. Prokaryote 16S rRNA gene amplicons for 2015 and 2016 4000 m trap time-series samples were generated in this study, and were prepared exactly as described in Boeuf et al.[28]. More specifically, the V4 region of the SSU rRNA gene was amplified using "universal"

primers 515F[45] (5′-GTGYCAGCMGCCGCGGTAA-3′) and 806RB[46] (5′-GGACTACNVGGGTWTCTAAT-3′). Amplicons were purified with AMPure XP beads (Beckman Coulter, Brea, CA, United States), and sequenced on an Illumina MiSeq platform as previously described in Li et al.[47].

For suspended microbial time-series samples at Station ALOHA, a total of 2L of whole seawater was collected from depths of 5–175 m, and a total of 4 L seawater was filtered from depths between 200–4000 m. Filtration by peristaltic pumping without prefiltration was performed using 0.2 um Supor filters (Pall Life Sciences, Port Washington, New York, United States), and samples were processed exactly as previously described[14]. Samples were collected on approximately a monthly basis on Hawaii Ocean Time-series cruises from Nov 2014 through Nov 2016 (Supplementary Data 6); https://hahana.soest.hawaii.edu/hot/hot-dogs/crssum.html).

PARAGON expedition (http://scope.soest.hawaii.edu/data/scope2021/) particle interceptor sediment traps (PITs)[1,8,48,49] were deployed and processed as previously described[49] with the following modifications: PIT traps were deployed at depths of 150, 175, 200, 250, 300 and 500 m during the PARAGON I cruise from July 23 – Aug 3, 2021 (Supplementary Data 7). The PIT traps were fitted with modified 250 mL collection jars capped with a funnel that formed a tight seal with the inside walls of the PIT trap. These modified collection jars were placed in the bottom of the PIT traps and filled with a full-strength saturated ammonium sulfate nucleic acid preservative[49]. Before deployment, the upper portion of each PIT trap (above the preservative-containing collection bottle) was filled with 0.2 um pre-filtered seawater adjusted with sodium chloride to a final density of 1.05 g/mL. Three identical PIT traps were deployed at each depth, for use as biological replicates. The PIT arrays were recovered after a 12-day drifting period. Following recovery, particles were filtered through a 335 um Nitex® mesh to remove larger zooplankton, and next filtered onto a 0.2 um Supor filter (Pall Life Sciences, Port Washington, New York, United States). Filters were placed in RNALater (Thermo Fisher Scientific, Waltham, MA) and stored at −80 °C until further processing. Suspended microbial seawater samples from the PARAGON cruise were collected and processed exactly as described for the Station ALOHA time-series water samples[14], and stored at −80 °C until processing (Supplementary Data 7). A total of 21 PARAGON sediment trap samples were analyzed for 16S rRNA gene ASVs, and a total of 9 water column depths from the PARAGON cruise were analyzed for 16S rRNA gene ASVs (Supplementary Data 7).

### DNA extraction and amplicon sequencing of water samples and PARAGON sediment trap samples

DNA extraction of the Station ALOHA time-series and PARAGON filtered seawater samples was performed as described in Leu et al.[14]. PARAGON sediment trap samples were extracted using the Power Biofilm Extraction Kit following manufacturer's instructions, as previously described[28] (Qiagen, Valencia, CA, United States). The 16S rRNA gene amplicon libraries were prepared as described above for Station ALOHA 4000 m trap samples. For all samples, the V4 region of the SSU rRNA gene was amplified using "universal" primers 515F[45] (5′-GTGYCAGCMGCCGCGGTAA-3′) and 806RB[46] (5′-GGACTACNVGGGTWTCTAAT-3′). Amplicons were purified with AMPure XP beads (Beckman Coulter, Brea, CA, United States) and sequenced on an Illumina MiSeq platform. All SSU rRNA amplicon sequences have been deposited in the NCBI Sequence Read Archive under project number PRJNA482655 (4000 m trap amplicons), PRJNA352737 (Station ALOHA suspended prokaryote amplicons), and PRJNA966198 (PARAGON expedition particle-associated and suspended prokaryote amplicons).

### Amplicon sequence processing

The SSU rRNA amplicons retrieved from water samples collected during Nov 2015 to April 2016 had lower quality scores at the 21st

base position in reverse reads. This nucleotide base was therefore masked from all reverse reads in subsequent amplicon processing. All reads were trimmed with Trimmomatic v0.36[50], and split into individual sample files using QIIME[51], as described in Li et al.[47]. The multiple sequencing runs were further processed individually with R package DADA2 (version 1.14.1)[52], as described in Li et al.[47]. Read trimming and error rate learning procedures were performed using DADA2 (version 1.14.1) functions using default settings, except for read length trimming, as follows. For PARAGON trap and water column sample amplicons, the forward reads were truncated after 100 bases and the reverse reads after 180 bases, due to lower quality scores after these positions, from this sequencing run. For all other samples, the forward reads were truncated after 180 bases and the reverse reads after 100 bases. To maximize identification of rare ASVs, samples were pooled together for downstream sequence variant analyses. After generating ASVs from individual sequencing runs, they were merged into a single, non-redundant ASV dataset using mergeSequenceTables() function in DADA2 with default settings. Chimeras were removed using the consensus method with defaults settings in DADA2. An ambiguous nucleotide ("N") was inserted to the 21st base position from the end of ASV sequence to account for poor sequence quality scores at this position. Taxonomic assignments used the IDTAXA algorithm[53] with the SILVA SSU r138 as the reference database[54].

### Sequence data analysis

Station ALOHA time-series 4000 m trap 16S rRNA gene amplicon ASVs were compared with Station ALOHA time-series water column 16S rRNA gene amplicon ASVs to obtain 4000 m trap SASVs (see the explanation below). Station ALOHA suspended water column ASVs were analyzed over depth and time, using averages calculated for each given month (Supplementary Fig. 1a). For example, March average relative abundance of water column ASVs at 5 m was the average value of March 2015 and 2016 at 5 m. September mean relative abundances of water column ASVs were not available for the whole water column while March mean values were not available at depths of 2000–4000 m, and July and November for 4000 m. The missing values for the monthly mean relative abundance of suspended water column ASVs were interpolated using the average of the two adjacent months. 4000 m trap SASVs were identified by comparing 4000 m trap-collected ASVs from individual time points with the corresponding monthly average of time-series water column ASVs from the same month. (For example, 4000 m trap SASVs at 5 m for 2014-3-27 were identified by comparing 4000 m trap-collected ASVs from 2014-3-27 with March average of water column time-series ASVs at 5 m). The 4000 m trap SASVs were obtained over all water column depths and across the 3-yr trap time series. The PARAGON 500 m sediment trap-collected ASVs were compared to corresponding PARAGON water column ASVs collected during the same time period, yielding the 500 m trap SASVs. The other PARAGON trap samples were processed similarly, to obtain the 300 m, 250 m, 225 m, 200 m, 175 m and 150 m trap SASVs.

The 4000 m trap SASVs that were ≥0.1% of their total time-averaged water column relative abundances for a given depth, were used to identify SASVs originating from specific depth horizons. This approach yielded 259 depth-stratified SASVs (Fig. 3a, Supplementary Data 3). Depth-averaged water column relative abundances of each SASV were then calculated (Fig. 3a, which shows the only dataset averaged by depth in our analyses). The depth maxima of these SASVs were obtained using find_peaks() in Python (version 3.8.1) using default settings. A total of 212 SASVs were identified as having a single depth maximum, while 47 SASVs had bimodal depth maxima. For these SASVs having bimodal peaks, 41 SASVs were retained for the following analyses only if: 1. Only one of the bimodal depth maxima exceeded the depth-averaged water column relative

abundance; or 2. Both depth maxima were present in a single depth zone, as categorized in Fig. 3. The other six SASVs with bimodal depth maxima were excluded from subsequent analyses. (These six SASVs having two depth peaks comprised 0.7–6.9% of the total SASV water column relative abundances at each depth, with a mean of 2.8%). The final 253 SASVs having one depth maximum were aggregated into six depth-zone specific categories (Fig. 3a) and were assigned to the most abundant taxa at each depth zone shown in Fig. 3.

### Power-law function determination for upper water column SASVs data

The average upper water column relative abundances of both the 4000 m and 500 m trap SASVs were used for simulating the power-law curve model. The best fit to a power-law function of the form $S_Z = S_{75} (Z/75)^b$ was observed for upper water column depths, using curve_fit() in Python (version 3.8.1) using default settings. The p value was evaluated using the method from zunzun.com curve fitting website with scipy.stats in Python (version 3.8.1) using default settings. 95% confidence intervals were calculated using two sigma uncertainties with unp.std_dev() in Python (version 3.8.1) using default settings. The results were similar to the Martin curve $F_Z = F_{100}$ $(Z/100)^b$ [8]. When the average water column relative abundance of 4000 m trap SASVs at a depth of 100 m was used as the log-log intercept ($S_Z = S_{100} (Z/100)^b$), the exponent $b$ was −0.88 and $R^2$ decreased to 0.68. A few studies previously suggested that net community productivity (NCP) was negligible in the mixed layer of the NPSG[55,56]. The average mixed layer during 2014–2016 was located at 59 m as estimated by a density model[57], and at 67 m by a temperature model[58] (https://hahana.soest.hawaii.edu/hot/hot-dogs/mldepth.html). In our study, we therefore selected 75 m as the reference depth of sinking particle export.

### SASV enrichment in 4000 m traps during the summer export pulse

Welch's t-test was employed to identify 4000 m trap SASVs that had significant differences in relative abundances in 4000 m traps during SEP vs. non-SEP time periods, using scipy.stats.ttest_ind() in Python (3.8.1). SASVs that had statistically significant greater average relative abundances in 4000 m traps during SEP compared to non-SEP time periods, were identified as enriched during SEP.

### Reporting summary

Further information on research design is available in the Nature Portfolio Reporting Summary linked to this article.

## Data availability

All 16S rRNA gene amplicon sequences have been deposited NCBIs Sequence Read Archive, under the following accession codes: All SSU rRNA amplicon sequences have been deposited in the NCBI Sequence Read Archive under project number PRJNA482655 (4000 m trap amplicons), PRJNA352737 (Station ALOHA suspended prokaryote amplicons), and PRJNA966198 (PARAGON expedition particle-associated and suspended prokaryote amplicons). All other metadata are available either in this report and associated Supplementary Data files, or in Boeuf et al.[28] and Poff et al.[13] (4000 m sediment trap data), Grabowski et al.[5] (attenuation curve data for POC, particle mass or particle energy content), or on the Station ALOHA Hawaii Ocean time-series website (water column oceanographic parameters for suspended ASV time-series collections).

## Code availability

All code used in this study that is not already publicly available, can be found at: https://github.com/fuyanli121/Codes_SASV-paper_2023 (https://doi.org/10.6084/m9.figshare.24328942).

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

## Acknowledgements

We thank the captains and crews of R/V Kilo Moana, R/V Ka'imikai-O-Kanaloa, the HOT program, and the SCOPE team for cruise organization, sample collection, and oceanographic data acquisition. We thank Tara Clemente, Blake Watkins, and Eric Grabowski for their expert efforts and leadership in sediment trap deployments, recoveries, and biogeochemical analyses. We are grateful to all our SCOPE colleagues who participated in the PARAGON I expedition in 2021. This research was supported by grants from the Simons Foundation (329108 to E.F.D. and D.M.K., 721223 to E.F.D., and 721252 to D.M.K.). This work is a contribution of the Simons Collaboration on Ocean Processes and Ecology and the Daniel K. Inouye Center for Microbial Oceanography: Research and Education.

## Author contributions

E.F.D. and D.M.K. conceived the study and supervised the work. F.L. and A.B. performed sample collection and fieldwork. A.B., F.L., and K.P. performed the laboratory work necessary for generating the rRNA gene amplicon sequences. F.L. and J.M.E. performed sequence analyses and statistical analyses. F.L. and E.F.D. wrote the manuscript, with contributions from D.M.K., A.B., J.M.E., and K.P.

## Competing interests

The authors declare no competing interests.
