## [Peer Review File · Nature Communications]

Planktonic microbial signatures of sinking particle export in the open ocean's interiorREVIEWER COMMENTS

Reviewer #1 (Remarks to the Author):

The submitted manuscript deals with an assessment of depth and temporal changes in prokaryotic communities in sinking and suspended particles. The work relates to assessments of the efficiency of the biological carbon pump, and the role of microbial communities in the sequestration of organic carbon in the deep ocean.

The work is novel and elegant, and provides new insights into the link between surface ocean and mesopelagic processes and carbon sequestration. The study uses the Hawaii time series site ALOHA for its assessment and hence is focused on the NPSG. The focus on a single specific region is mentioned as a shortcoming in the final section of the paper, where it is mentioned that further studies are required elsewhere; this is correct.

Overall, I see this as an excellent contribution to Nature Communications. An improved microbial assessment of the export of organic matter, production of organic matter in the mesopelagic and the delivery of material (and prokaryotes) to the deep ocean, are very important to our assessment of the functioning of the biological carbon pump. Hence this study is important and unique.

There is a strong influence of temperature on the remineralisation of organic matter in the top of the water column, reflected in the B coefficient. The authors may want to comment on this, as it provides another dimension to their limited geographical remit of the study.

The manuscript also requires further quantification in the text of some of the processes. A range of issues are in the SI, but quantification will need to be presented in the main text. The b coefficient of the Martin curve needs to be explained better. The relevance of the change in microbial abundances along the Martin trend needs to be explained and interpreted.

My expertise is not in genomics but in the carbon pump, and I will not assess specific details of the genomics part of the work. I hence cannot assess the genomics methodology of the work. All other methodologies are sound.

The manuscript contains a host of grammatical errors, which need to be addressed. I mentioned a number below.

I recommend publication of the manuscript following moderate revision.

Specific comments.

Line 22 and 39. The term 'considerable' is used here. It will be appropriate to quantify this, which is best done by provide a range.

Line 23. It should be carbon

L 25 oceans' or ocean's??? I suppose you mean the second one.

L 28the water column relative abundance of suspended prokaryote taxa..... too many adjectives (also line 81).

L 39: (carbon dioxide) CO₂ should be carbon dioxide (CO₂)

L 40; photosynthetically derived biomass along with non-living carbon.....not clear what is meant here. Sinking organic matter is typically derived from phytoplankton debris (and it is non-living).

Fig. S2. What are the energy curves? Not explained.....

Table S1 and S3. What are STAVSs? What is SEP?

Line 90: The flux attenuation coefficient is mentioned here (and further on in the manuscript). The authors do not mention any values of b in this study. This is confusing. B is typically treated as a variable, with the value changing with the strength of the organic matter remineralisation in different water column depth horizons.

Line 100: why would we really need another indicator of depth dependent POC attenuation in the form of microbes? I would think that the observed shape of the attenuation profile of the microbes is interesting and worth interpreting, but they are perhaps a better reflection of food availability.

L 118: what is ATP here?

L 173: energy? Where does this statement come from?

L 176: The export of smaller suspended picophytoplankton.... This phrase contradicts itself.

Reviewer #2 (Remarks to the Author):

The manuscript provides a detailed 16S rRNA gene amplicon survey of microorganisms throughout the water column and sediment/particle traps at Station ALOHA over multiple time series sampling efforts. The data are used to compare water column and particle-entrained taxa across multiple different depths to identify organisms that are common between both communities and note the degree of decay or enrichment of these shared taxa with depth. The results demonstrate that many organisms can serve "as reliable tracers of particle export". The study relies on an unusually extensive dataset through both space and time, and the conclusions are plausible (although I note some statistical issues below). However, I'm not sure whether the impact of the science is sufficient for Nature Communications. I'll let the editor decide based on my comments below and those of the other reviewers. Regardless of whether this is accepted here or elsewhere, I believe there are several issues that could be addressed to make the paper stronger:

General

1. There is a lot of averaging done throughout the paper (depth averaging, time averaging, replicate averaging) and yet there are almost no examples of any ranges being reported to allow the reader a sense of the variability of the data. I'm also not sure the appropriate statistical examination of the data was conducted either. Most of the conclusions hinge on this point, because the curves in figure 1 and S2 are only worth reporting if x-axis values are significantly different from each other at each datapoint. Figure 1e is the only example where this kind of data-range is reported, and it's only for one of the three curves on the plot. It's suspicious that this was done for one curve but none of the others. I would like to see 95% confidence intervals for all the curves and a statistical evaluation of whether the decay is significant. This problem also extends to the heat plots, which are exclusively qualitative. I would like to see quantitative evaluation of the differences in ASVs with depth, season, etc. Given the extensive replication in the dataset (which is amazing and very unusual), there is a unique opportunity here evaluate the significance of the results and really prove the point.

2. The methods description suffers from some very wordy and unclear sentences, and the figure S1 designed to explain the experimental design did not help me understand the overall sampling and analysis scheme. I note specific places where confusion arose for me below. Versions and settings (including flags, etc.) for all software need to be provided, and the scripts used in the analysis need to be made publicly available too.

3. I wasn't convinced that the utility of this method was a major improvement on what we would already know from just studying POM dynamics alone, especially given the cost of the method and the salaries of the people to collect and analyze the data. For example, on L196, the statement is made that "suspended microbes entrained in sinking particles can provide depth-specific details about sinking particle origins and export dynamics." This is a great claim, but I see several problems with it. First, I don't see what we actually learned about sinking particle origins. They come from the surface! We don't need to measure surface 16s rRNA gene sequences in the particle to determine that. Although the data indicate enrichment of organisms in the deeper samples, and this was suggested to result from "potentially, increased deep-sea new particle formation" (L188), there is no POM concentration data to back up such a claim, so it can't be evaluated. Second, what export dynamics are they referring to? No degradation or production measurements were included in the study, nor were seasonal variations in export. The only thing that was measured was changes in microbial communities with 16S, but this doesn't tell us about particle dynamics, only about microbial community dynamics, and even then, the community membership hasn't been linked to microbiological function. These kinds of claims are repeated in multiple places, but they are pretty vague and it wasn't clear to me what we've really learned other than how microbial abundance varies on particles by depth. The authors would do well to be much more precise in their language and emphasize clearly how 16S rRNA gene amplicons give us significantly new information about particle export dynamics that we wouldn't get from other approaches.

Specific

L65 and throughout- please ensure all instances of "16S rRNA" are changed to "16S rRNA gene".

L68-70: This isn't really a falsifiable hypothesis, is it. Simply having "can" in the hypotheses invalidates it, but also having such vague and non-directional words like "details" and "processes" means that there's no way to evaluate the predictions of the hypothesis, and therefore to falsify it. Please omit this sentence in favor of stating a research objective, or re-make the hypothesis such that it makes specific predictions, and then of course, evaluate whether those predictions have been met.

L82 is an example of my general issue #1, where means are reported without ranges, so we have no way of knowing if the differences are significant. Other examples are L114, 116, all the main figures, and more.

Please provide references for the sentence ending on L108.

Please provide a reference for the claim of enhanced degradation rates on L109.

L99-111: Please be more precise- what signatures? What particle dynamics? What is revealed by studying 16S rRNA gene amplicons that we don't already know about particle export?

L143-145: This section would be a lot more compelling if the authors explored what's known about the microbes they've observed and what this might mean for degradation/production of particles. For example, many of the organisms they're discussing are known traditionally as "free-living" taxa (and many are non-motile). Thus, their presence on sinking particles either means that we're very wrong about their normal lifestyles, or they're getting collected like bugs on a windshield. The converse is true of course for organisms known to be associated with particles. The whole paper would benefit from a more thorough discussion of the microbiology behind the 16S rRNA gene sequences.

L159: Why are these three genera receiving extra focus? It's not clear why this paragraph highlights just these three.

L169: HOW might copiotrophs "influence sinking particles" (and what do you mean by influence)?

L174: By "Here" do you mean Station ALOHA?

L174-175: Please provide references for this sentence.

L184-185: Which taxa are in these different categories? And why are chemolithotrophic taxa considered independent of the oligotroph/copiotroph spectrum?

L188: Is there evidence of new particle formation in any POM data from anywhere? Please cite references.

L210: Is there any metadata for these samples? E.g., POM/DOM concentrations, temperature, salinity, nutrients, etc. Has that already been reported elsewhere? Is there a reason that wasn't used here to provide more details about what's changing from a marine chemistry perspective?

L226: What kind of filtration is this referring to? L229 says no pre filtration...

L229-230: Is the sampling schedule detailed somewhere?

L231-235: How many stations/timepoints were involved in this sampling effort?

L273-275: Why were the reads truncated differently depending on the samples?

L277-278: How were the ASVs merged? Usually you need them all together in one batch during the analysis so you don't have the same ASV called two different things.

L282: How were replicates treated in the analysis?

L285: For what parts of the analysis were samples averaged by depth and/or time? And why? Why not explore the temporal or spatial resolution more completely? Also, why aren't ranges provided after averaging?

L292-294: Why were individual time points from 4000m traps compared to monthly averages of water column ASV abundances and not the same time points? To account for sinking lag times? Were the sample dates not coordinated?

L297: This 500 m trap is from the Paragon cruise, right?

L306-308: I read this sentence at least 4 times and I'm not totally sure I got it. Please re-word.

L313-317: These sentences seem non-sequitur. Why the sudden emphasis on intra- vs. extracellular DNA?

Was the Paragon data combined with the other ALOHA data, or were these kept separate for different analyses? Please detail how they were used together or separately.

RESPONSES TO REVIEWER COMMENTS

Reviewer #1 (Remarks to the Author):

The submitted manuscript deals with an assessment of depth and temporal changes in prokaryotic communities in sinking and suspended particles. The work relates to assessments of the efficiency of the biological carbon pump, and the role of microbial communities in the sequestration of organic carbon in the deep ocean.

The work is novel and elegant, and provides new insights into the link between surface ocean and mesopelagic processes and carbon sequestration. The study uses the Hawaii time series site ALOHA for its assessment and hence is focused on the NPSG. The focus on a single specific region is mentioned as a shortcoming in the final section of the paper, where it is mentioned that further studies are required elsewhere; this is correct.

Overall, I see this as an excellent contribution to Nature Communications. An improved microbial assessment of the export of organic matter, production of organic matter in the mesopelagic and the delivery of material (and prokaryotes) to the deep ocean, are very important to our assessment of the functioning of the biological carbon pump. Hence this study is important and unique.

Thank you for these positive comments on our manuscript, and the constructive criticisms and suggestions that follow below. We believe your comments, and our subsequent responses and revisions, are helping to improve the quality and clarity of the revised manuscript.

There is a strong influence of temperature on the remineralisation of organic matter in the top of the water column, reflected in the B coefficient. The authors may want to comment on this, as it provides another dimension to their limited geographical remit of the study.

We have examined our datasets further, done some additional analyses, and added considerations of environmental parameters in the revision. This includes a short discussion on temperature on remineralization in lines 131-143 in the revision, and additional statistical analyses on the relationship between environmental parameters and the Martin curve, as indicated in Supplementary Fig. 3c. The bottom line is that compared to temperature, elemental nutrient concentrations, such as phosphate, nitrite+nitrate, and silicate, exhibited the greatest negative correlations with b coefficients within and below DCM. We believe this most likely reflects the greater remineralization of particulate organic matter and thus the enhanced concentration of dissolved inorganic nutrients in the upper water column (reference 17). In the revision (line 142-143) we state: “Significant correlations between temperature and Martin curve b coefficients were not observed, likely because most particle degradation in the NPSG occurs in the warm upper water column.”

The manuscript also requires further quantification in the text of some of the processes. A range of issues are in the SI, but quantification will need to be presented in the main text. The b coefficient of the Martin curve needs to be explained better. The relevance of the change in microbial abundances along the Martin trend needs to be explained and interpreted.

As suggested, we have included more explicit quantitative information in the revision, and more detail on analyses and information in the SI. Please see lines 91-97, 100-102, 148-154, 190-197 in the revision, where some of this additional information and explanation appears.

On lines 46-52, we have also expanded our explanation of the Martin curve and b coefficient as recommended, as follows: “In the upper water column, the POC flux decreases exponentially with depth. One commonly used empirical approach to describe particle flux attenuation with depth is known as the “Martin curve”⁸. The model is expressed as $F = F100(z/100)^{-b}$, where z is the POC collection depth, F100 is the reference POC flux at 100 m, and b is a unitless parameter that represents the magnitude of flux attenuation with depth. The Martin curve power-law relationship is typically used to describe POC flux attenuation between depths of ~100 – 2000 m⁸, where sinking particle degradation typically exceeds its formation.”

My expertise is not in genomics but in the carbon pump, and I will not assess specific details of the genomics part of the work. I hence cannot assess the genomics methodology of the work. All other methodologies are sound. The manuscript contains a host of grammatical errors, which need to be addressed. I mentioned a number below.

We have done our best to correct any grammatical errors throughout the revision and in response to Reviewer 1’s specific comments, as indicated below:

I recommend publication of the manuscript following moderate revision.

Thank you.

Specific comments.

Line 22 and 39. The term ‘considerable’ is used here. It will be appropriate to quantify this, which is best done by provide a range.

Thanks for the suggestion. We have now included ranges in the text suggested and as appropriate, as follows:

Line 23: A considerable amount of particulate carbon produced by oceanic photosynthesis is exported to the deep-sea by the “gravitational pump” (~6.8 to 7.7 Pg C/year) , sequestering it from the atmosphere for centuries¹⁻⁴.

Line 40: Globally, a considerable fraction (10-18%) of this net primary productivity is transported to the deep sea via sinking particles where it nourishes the deep-sea ecosystem and is sequestered from the upper ocean for centuries⁴ .

Line 23. It should be carbon

Thank you – now corrected in the revision, with the ranges cited from Nowicki et al., reference 4, as above in lines 23 and 40.

L 25 oceans' or ocean's??? I suppose you mean the second one.

The abstract has been revised and this term no longer appears.

L 28the water column relative abundance of suspended prokaryote taxa..... too many adjectives (also line 81).

We do apologize for the wordiness, but while somewhat awkward, it is necessary. This is due to the somewhat complex nature of the calculated values, which requires being specific – all those adjectives have specific meaning that is required for the reader to properly interpret our analyses and conclusions.

L 39: (carbon dioxide) CO2 should be carbon dioxide (CO2)

Corrected in the revision.

L 40; photosynthetically derived biomass along with non-living carbon.....not clear what is meant here. Sinking organic matter is typically derived from phytoplankton debris (and it is non-living).

To avoid any potential confusion, this sentence has now been changed in the revision to: “Globally, a considerable fraction (10-18%) of this net primary productivity is transported to the deep sea via sinking particles where it nourishes the deep-sea ecosystem and is sequestered from the upper ocean for centuries⁴ .”

Fig. S2. What are the energy curves? Not explained.....

We have further elaborated on the specifics of the enthalpy curves (specific energy content of particles in Joules per unit mass) derived from Grabowski et al. (2019) in the text (lines 115-116), and in the Fig. S2.

Table S1 and S3. What are STAVSs? What is SEP?

The “STAVS” term was included as an error, and was removed in the revision, and replaced with SASVs where appropriate (e.g., Table S1 and S4).

SEP refers to the well documented “summer export pulse” (SEP) that occurs regularly in the NPSG (from Karl et al. 2012, reference 7), and defined as those time periods when particulate nitrogen and particulate carbon flux measured at 4000 m, exceed annual mean fluxes by 150% or more. This is further elaborated on in the main text in Line 218 in the revision. We also included this definition in Table S1 and S4.

Line 90: The flux attenuation coefficient is mentioned here (and further on in the manuscript). The authors do not mention any values of b in this study. This is confusing. B is typically treated as a variable, with the value changing with the strength of the organic matter remineralisation in different water column depth horizons.

We have added more discussion in the revision on the variation of b coefficients, which we observed. Please see especially our expanded explanation of the Martin curve on lines 46-52, and lines 131-143 in the revision.

Line 100: why would we really need another indicator of depth dependent POC attenuation in the form of microbes? I would think that the observed shape of the attenuation profile of the microbes is interesting and worth interpreting, but they are perhaps a better reflection of food availability.

We agree understand your question here, and have revised and expanded the text accordingly, with regard to our central aim. We state explicitly (lines 112-113) : “Can quantitative data on suspended microbial entrainment in sinking particles provide useful information on the microbial dynamics and biology of POC export?” While the patterns we observe may indeed reflect food availability (amongst other things), the observation that this very different data type recapitulates known patterns (e.g. the Martin curve), supports its use and interpretation here. This further suggests these data can also provide new information and biological perspective on particle degradation and formation throughout the water column. We expand further on these ideas in the revised paragraph (lines 112- 130).

L 118: what is ATP here?

ATP is adenosine triphosphate, and is now explicitly defined in the revision.

L 173: energy? Where does this statement come from?

We have further elaborated on the specifics of the enthalpy curves (specific energy content of particles in Joules per unit mass) derived from Grabowski et al. (2019), reference 5, in the text (lines 115-116), and in the Fig. S2 legend.

L 176: The export of smaller suspended picophytoplankton.... This phrase contradicts itself.

We deleted the terms “suspended” in this sentence in the revision.

Reviewer #2 (Remarks to the Author):

The manuscript provides a detailed 16S rRNA gene amplicon survey of microorganisms throughout the water column and sediment/particle traps at Station ALOHA over multiple time series sampling efforts. The data are used to compare water column and particle-entrained taxa across multiple different depths to identify organisms that are common between both communities and note the degree of decay or enrichment of these shared taxa with depth. The results demonstrate that many organisms can serve "as reliable tracers of particle export". The study relies on an unusually extensive dataset through both space and time, and the conclusions are plausible (although I note some statistical issues below). However, I'm not sure whether the impact of the science is sufficient for Nature Communications. I'll let the editor decide based on my comments below and those of the other reviewers. Regardless of whether this is accepted here or elsewhere, I believe there are several issues that could be addressed to make the paper stronger:

Thank you for your comments and suggestions, that we believe are helping to improve the clarity and conclusions in the revised manuscript. We have attended to the statistical issues mentioned by Reviewer 2, and further refined the revision to avoid any overstatements or misstatements. We also further discuss the novelty, utility and impact of our approach for improving especially our biological understanding the sinking particle export in the ocean. We believe that our manuscript provides a unique and informative perspective on sinking particle export, and we try to emphasize this more, and why, in the revision.

General

- 1. There is a lot of averaging done throughout the paper (depth averaging, time averaging, replicate averaging) and yet there are almost no examples of any ranges being reported to allow the reader a sense of the variability of the data. I'm also not sure the appropriate statistical examination of the data was conducted either. Most of the conclusions hinge on this point, because the curves in figure 1 and S2 are only worth reporting if x-axis values are significantly different from each other at each datapoint. Figure 1e is the only example where this kind of data-range is reported, and it's only for one of the three curves on the plot. It's suspicious that this was done for one curve but none of the others. I would like to see 95% confidence intervals for all the curves and a statistical*

evaluation of whether the decay is significant. This problem also extends to the heat plots, which are exclusively qualitative. I would like to see quantitative evaluation of the differences in ASVs with depth, season, etc. Given the extensive replication in the dataset (which is amazing and very unusual), there is a unique opportunity here evaluate the significance of the results and really prove the point.

We thank Reviewer 2 for these comments, and we agree that where possible, the averages, ranges and/or confidence intervals can and should be shown. We have included this information now in both the text and the Figures in the revision. Please see Figs. 1c, 1d, 1e, 4c, Supplementary Tables 1, 2, 4 and Supplementary Fig. 2a, 3a, 3b, 4a, 4b, and text lines 91-97, 100-102, 148-154, 190-197, where these suggested (and useful, thank you) revisions can be found.

All power-law curves from individual time points and replicates reported in our study are shown in Supplementary Table 2 and Supplementary Fig. 3a, 4a, and their 95% confidence levels are supplied in Supplementary Table 2 and Supplementary Fig. 3b, 4b. Statistical p values are also evaluated and included in Supplementary Table 2. All the decay curves were statistically significant.

The depth-specific decrease in upper water column relative abundance of SASVs is fitted the best to power-law curve which is statistically significant for each time point and each replicate. The depth-specific increase in both water column and trap relative abundances of SASVs in deep water were shown in Fig. 1e and 4c (both with standard deviation). This additional quantitative information has been included in Supplementary Table 1, 4 and text lines 91-97, 100-102, 148-154, 190-197, in the revision.

In the original manuscript we did briefly describe a quantitative evaluation of taxa that varied significantly with time and season (currently in Supplementary Table 5 in then revision), but this was not extensively discussed. In Fig. 1a, in terms of the SASV water column relative abundances, we did not observe significant difference between seasons. In Fig. 4a, the SASV trap relative abundances did exhibit significant differences between the SEP compared to non-SEP time periods. Those taxa that were found to be significantly enriched ($p < 0.05$, see Table S5) seasonally (e.g., SEP versus non-SEP) are shown in Supplementary Table 5. In the revision, we discuss in more detail these seasonal differences. Please see text lines 215-231, for where these revisions occur.

2. The methods description suffers from some very wordy and unclear sentences, and the figure S1 designed to explain the experimental design did not help me understand the overall sampling and analysis scheme. I note specific places where confusion arose for me below. Versions and settings (including flags, etc.) for all software need to be provided, and the scripts used in the analysis need to be made publicly available too.

We thank Reviewer 2 for these comments, and we have attempted to clarify and refine the Methods text in the revision. Please also see our responses below for how we address each of Reviewer 2's questions and concerns in the revision.

- 2. I wasn't convinced that the utility of this method was a major improvement on what we would already know from just studying POM dynamics alone, especially given the cost of the method and the salaries of the people to collect and analyze the data. For example, on L196, the statement is made that "suspended microbes entrained in sinking particles can provide depth-specific details about sinking particle origins and export dynamics." This is a great claim, but I see several problems with it. First, I don't see what we actually learned about sinking particle origins. They come from the surface! We don't need to measure surface 16s rRNA gene sequences in the particle to determine that.*

We thank Reviewer 2 for these comments. We have attempted, throughout the revision, to avoid any overstatement, and to be more precise in our conclusions.

The SASV relative abundance patterns and depth-specific suspended microbial community compositions provide new information on probable particle origins, and therefore export dynamics, based on the biology and ecology. That particles derive from photosynthesis originally formed in the surface is well known, and this has been validated in many studies. However, previous studies have also shown that particles may be produced in deeper waters (Karl and Kauer, *Oceanographic Research Papers*, 1984; Karl and Kauer, *Bulletin of Marine Science*, 1984; Karl et al., *Nature*, 1984). Not all particles at depth necessarily originate directly from the surface – some are processed, transformed and repackaged as they sink. There is in essence, a “bucket brigade” of particle formation, downward flux, consumption, repackaging, further downward flux, and so on, that occurs as particles traverse from the surface to the seafloor.

Our approach provides new perspective on some of the details of depth-dependent processes of particle transformations during downward transit. For example, our approach recapitulated known depth-dependent attenuation patterns of particle flux in the upper water column, despite using a completely different data type (SASVs). We also see different patterns in the mesopelagic and deeper, which are new informative and have not been previously reported.

In the revised manuscript we are more specific about novel findings (at the same time trying not to overstate our case), throughout. For example, please see revised lines 112-130.

“Can quantitative data on suspended microbial entrainment in sinking particles provide useful information on the dynamics and biology of POC export? Although the methodologies we used were quite different from prior studies, the SASV attenuation curves recapitulated depth-dependent trends that have been previously reported for chlorophyll, POC, energy (enthalpy, or specific energy

content of particles in Joules per unit mass), and mass depth attenuation in the NPSG^{1, 5} (Fig. 2). Notably, the biogeochemical data we used here from Grabowski et al.⁵ (Fig. 2), were collected six months prior to the 4000 m trap SASV time-series data we report here from the same study site. The trends observed were consistent for time-averaged data collected over three years (in sediment traps deployed at 4000 m), as well as those deployed for shorter time periods at a variety of shallower depths (175, 200, 250, 300 and 500 m) (Fig. 2, Supplementary Fig. 2). A notable difference between the deep and shallow traps however, is that the attenuation of SASVs in the upper water column appear slightly more pronounced in shallow traps compared to deep traps (Fig. 2, Supplementary Fig. 2). The lesser contribution of upper water column ASVs to the 4000 m deep trap SASVs, (Fig. 1c-e), likely reflects the elevated degradation of particles that sink to abyssal depths^{5, 7}. These analyses are consistent with the known trend of sinking particle consumption exceeding its production at these depths at Station ALOHA (75- 250 m)^{5, 7}. These results in total support the proposal that analyses of suspended microbes that are entrained in sinking particles at different depths, can yield useful information that is relevant to sinking particle degradation, formation and biotic processes.”;

Also please see lines 246-257: “In the mesopelagic and bathypelagic zones, the proportions of suspended prokaryotes entrained in sinking particles (including heterotrophic oligotrophs and copiotrophs, and chemolithotrophs) generally increased with increasing depth. The taxonomic richness of SASVs reaching 4000 m also increased, the greater the depth of SASV origin in the water column (Supplementary Fig. 8). These trends suggest increasing prokaryote entrainment, colonization and growth on sinking particles with depth, and increased new sinking particle formation in the deep-sea^{20, 21, 23}. Some of the biological and ecological insights derived from such analyses cannot be derived from biogeochemical data alone.”

We have attempted to improve clarity and accuracy of such statements throughout the revision.

Although the data indicate enrichment of organisms in the deeper samples, and this was suggested to result from "potentially, increased deep-sea new particle formation" (L188), there is no POM concentration data to back up such a claim, so it can't be evaluated. Second, what export dynamics are they referring to? No degradation or production measurements were included in the study, nor were seasonal variations in export. The only thing that was measured was changes in microbial communities with 16S, but this doesn't tell us about particle dynamics, only about microbial community dynamics, and even then, the community membership hasn't been linked to microbiological function. These kinds of claims are repeated in multiple places, but they are pretty vague and it wasn't clear to me what we've really learned other than how microbial abundance varies on particles by depth. The authors would do well to be much more precise in their language and emphasize clearly how 16S rRNA gene

amplicons give us significantly new information about particle export dynamics that we wouldn't get from other approaches.

There are POC and PON data included in our analyses, both original and revised – please see Figure 4a, which specifically reports time series data of POC and PON flux reaching 4000 m, that is synoptic with our deep trap SASV analyses. Indeed, our data do indicate that along with increased flux from surface “blooms” during the SEP, there is a deep-sea elevated response in this time period, that does indicate *“potentially, increased deep-sea new particle formation”*. We also now include analyses that related the upper water attenuation curve to measured biogeochemical parameters – please see Supplementary Fig. 3 and associated discussion in the revision (lines 138 -143).

Time-series studies of microbial communities associated with sinking particles coupled with synoptic time-series data of suspended microbial communities, have not been previously reported to our knowledge. This represents one of the novel aspects of our study. Our manuscript provides a unique perspective in particular on previously unreported patterns of increased suspended microbial entrainment in sinking particles in the upper mesopelagic, which eventually decays and levels off at greater depths. The increased contribution of deep water suspended prokaryotes and prokaryotic colonizers to the sinking particles in the mesopelagic, that eventually sink to 4000 m, implies both greater levels of particle formation at mesopelagic depths (more suspended oligotrophic taxa are “trapped” in particles), as well as elevated levels of microbial colonization in deeper waters (increasing numbers of copiotrophic colonizers). In total, our study provides new perspective on the biology, ecology and biodynamics of sinking particle export, that cannot be derived from biogeochemical data alone. Indeed, as shown in Figure 2 and elsewhere, our analyses quite complementary to such datasets, and add a new (micro)biological layer to our understanding of particle export to the deep sea. Future studies are needed to constrain the particle export and carbon sequestration processes, as discussed in lines 258-272 in the revision.

We appreciate Reviewer 2's constructive comments here, and have attempted to improve the clarity and accuracy of our conclusions throughout the revision. We also have added subtitles for each section in the Results and Discussion, that we hope improves the clarity and flow of the text.

Specific

L65 and throughout- please ensure all instances of "16S rRNA" are changed to "16S rRNA gene".

Thanks. We have corrected this throughout the revised manuscript.

L68-70: This isn't really a falsifiable hypothesis, is it. Simply having "can" in the hypotheses invalidates it, but also having such vague and non-directional words like

"details" and "processes" means that there's no way to evaluate the predictions of the hypothesis, and therefore to falsify it. Please omit this sentence in favor of stating a research objective, or re-make the hypothesis such that it makes specific predictions, and then of course, evaluate whether those predictions have been met.

We agree. We have removed “hypothesis” here and throughout the manuscript and replaced it with more appropriate terms. The above sentence now reads: “Here, we aim to determine whether suspended prokaryotes once entrained in sinking particles, can provide information about depth-specific particle export processes throughout the water column”

L82 is an example of my general issue #1, where means are reported without ranges, so we have no way of knowing if the differences are significant. Other examples are L114, 116, all the main figures, and more.

We have now included more quantitative information and statistical analyses in the revision. Please see lines 91-97, 100-102, 148-154, 190-197, as well as the standard deviation bars in Fig. 1c, 1d, 1e, 4c and Supplementary Fig. 2a in the revised manuscript.

Please provide references for the sentence ending on L108.

Please provide a reference for the claim of enhanced degradation rates on L109.

We added references for the statements of these two sentences in the revision. Please see the lines 126 and 128 in the revision.

L99-111: Please be more precise- what signatures? What particle dynamics? What is revealed by studying 16S rRNA gene amplicons that we don't already know about particle export?

We have attempted to be more explicit and clearer regarding these points in the revision. Please see lines 128-130, 246-253 in the revised text. One major finding of our study not evident from prior work are the differences in entrained microbes as a function of depth. And the large increase in SASVs that occurs in and below the upper mesopelagic. This appears to reflect both an increase in entrapment of suspended microbes (well documented oligotrophs that are stratified in the water column) at specific depths, in addition to in situ colonization as evidence by the increase of known deep water copiotrophic microbiota associated with particles at different depths. Our observations provide new clues about particle degradation, formation, and colonization throughout the water column which as of yet, are not well constrained especially in the mesopelagic and deeper.

L143-145: This section would be a lot more compelling if the authors explored what's known about the microbes they've observed and what this might mean for degradation/production of particles. For example, many of the organisms they're

discussing are known traditionally as "free-living" taxa (and many are non-motile). Thus, their presence on sinking particles either means that we're very wrong about their normal lifestyles, or they're getting collected like bugs on a windshield. The converse is true of course for organisms known to be associated with particles. The whole paper would benefit from a more thorough discussion of the microbiology behind the 16S rRNA gene sequences.

Thanks for this comment. The answer appears to be somewhere in the middle – that is, unicellular, planktonic oligotrophs do appear to be passively entrained in particles (for example Prochlorococcus and Pelagibacter from the upper water column). And there is also evidence for copiotrophic colonization/growth for specific taxa known to have particle-associated lifestyles and adaptations, at mesopelagic depths. This is now more clearly pointed out in the revision, and we discuss these issues detail throughout the text – see in particular, lines 167-179, 209-213 and 246-253.

L159: Why are these three genera receiving extra focus? It's not clear why this paragraph highlights just these three.

These three genera were the most abundant SASVs observed in 4000 m traps (Fig. 4d), hence the additional focus. The deeper water ecotypes of these mostly copiotrophic genera are very rare in the water column (categorized as 'Other' in Fig. 3b). They can colonize particles, grow there, and thus contribute significantly to the 4000 m trap particles community composition, compared to free-living prokaryotes which are entrained on particles mostly passively, and typically grow under very low nutrient, oligotrophic conditions. We clarified this in text lines 209-213 in the revision.

L169: HOW might copiotrophs "influence sinking particles" (and what do you mean by influence)?

These copiotrophs colonize particles, as well as grow on particles by degradation of organic matter. They are also known to encode a variety of exoenzymes that degrade high molecular weight proteins, carbohydrates and nucleic acids, thereby converting POM to DOM (see here especially refs 14 & 16). This is further elaborated in the revision on lines 209-213.

L174: By "Here" do you mean Station ALOHA?

Correct. We changed 'here' to 'At Station ALOHA'.

L174-175: Please provide references for this sentence.

Appropriate references have been added (line 238) in the revision.

L184-185: Which taxa are in these different categories? And why are chemolithotrophic taxa considered independent of the oligotroph/copiotroph spectrum?

We have retained the term chemolithotrophic here, given their physiological distinction from heterotrophic oligotrophs and copiotrophs. To make this clearer we revised the text to now read: “In the mesopelagic and bathypelagic zones, the proportions of suspended prokaryotes entrained in sinking particles (including heterotrophic oligotrophs and copiotrophs, and chemolithotrophs) generally increased with increasing depth.” The particular taxa involved can be found in Fig S6, Fig 3, 4 legend, and Table S3. We have added some more explicit references to various taxa associated with these general categories, in the revision.

L188: Is there evidence of new particle formation in any POM data from anywhere? Please cite references.

Appropriate references have now been added (line 252) in the revision.

L210: Is there any metadata for these samples? E.g., POM/DOM concentrations, temperature, salinity, nutrients, etc. Has that already been reported elsewhere? Is there a reason that wasn't used here to provide more details about what's changing from a marine chemistry perspective?

Oceanographic data was indeed used and is available for water column environmental parameters from <https://hahana.soest.hawaii.edu/hot/hot-dogs/>. In the revision, we have also included Tables in the supplement that describe sampling schedule and metadata associated with the water column time-series (Supplementary Table 6), and the sampling schedule for the PARAGON cruise (Supplementary Table 7). We use these data to test the correlations between environmental parameters and b coefficients for Station ALOHA time-series SASV power-law curves. Please see text lines 131-143 in revision. Further, as mentioned above, 4000 m trap-derived POC and PON data were also included in our time-series analyses, and Figures and discussion (and associated references in the text). Please see especially Figure 4a, and relevant discussion and references that appear in the text.

L226: What kind of filtration is this referring to? L229 says no pre filtration...

To be clearer, we changed ‘filtered’ to ‘collected’ in the revision. No pre-filtration was used.

L229-230: Is the sampling schedule detailed somewhere?

All suspended microbial samples (except for those collected on the PARAGON cruise), were collected on Hawaii Ocean Time-series cruises. The cruise information can be found here: <https://hahana.soest.hawaii.edu/hot/hot->

dogs/crssum.html. Specific cruise information is provided on line 296 in the revision, and we have included a data table (Supplementary Table 6) that explicitly includes the relevant information on suspended microbial collection, metadata, etc. We also now include the sampling schedule for PARAGON cruise as well, in Supplementary Table 7 in the revision.

L231-235: How many stations/timepoints were involved in this sampling effort?

For the PARAGON cruise, all sediment traps were deployed in at a single station, with each floating trap array sampling six depths (please see the text lines 298-301 and its cited references). As stated in the methods, the PARAGON sediment traps were deployed on July 23, 2021 on PARAGON cruise (<http://scope.soest.hawaii.edu/data/scope2021/>), and recovered after 12-day drifting period. We further clarified this in text lines 301 and 308. We also now include the sampling schedule for PARAGON samples analyzed here in Supplementary Table 7 of the revision. A total of 21 PARAGON sediment trap samples were analyzed for 16S rRNA gene ASVs, and a total of 9 water column depths from the PARAGON cruise were analyzed for 16S rRNA gene ASVs. We have added this information to the Methods section for greater clarity.

L273-275: Why were the reads truncated differently depending on the samples?

This was due to the fact that sequences from PARAGON samples yielded forward DNA sequence reads having lower quality scores after 100 bases, while reverse reads had lower quality scores after 180 bases. We have provided further detail on sequence processing relevant to this question in lines 341-351 in the revision, as follows: “Read trimming and error rate learning procedures were performed using DADA2 (version 1.14.1) functions using default settings, except for read length trimming, as follows. For PARAGON trap and water column sample amplicons, the forward reads were truncated after 100 bases and the reverse reads after 180 bases, due to lower quality scores after these positions, from this sequencing run. For all other samples, the forward reads were truncated after 180 bases and the reverse reads after 100 bases. To maximize identification of rare ASVs, samples were pooled together for downstream sequence variant analyses. After generating ASVs from individual sequencing runs, they were merged into a single, non-redundant ASV dataset using mergeSequenceTables() function in DADA2 with default settings. Chimeras were removed using the consensus method with defaults settings in DADA2.”.

L277-278: How were the ASVs merged? Usually you need them all together in one batch during the analysis so you don't have the same ASV called two different things.

After generating ASV sequence table from individual sequencing runs, they were merged into the single ASV dataset used in our study with mergeSequenceTables() in DADA2 using default settings, to eliminate

redundancy and combine identical ASVs from different sequencing runs into a single ASV. We added more detail on use of this function as described above.

L282: How were replicates treated in the analysis?

The biological replicates of PIT traps were processed together in DADA2.

L285: For what parts of the analysis were samples averaged by depth and/or time? And why? Why not explore the temporal or spatial resolution more completely? Also, why aren't ranges provided after averaging?

Only Fig 3a has SASV abundances averaged by depth. We clarified this in text line 380 in the revision. With time-averaged relative abundance data, the Fig. 3a heatmap would only visualize the most abundant taxa (red color), however, wouldn't be able to show the depth peak of each taxon.

The temporal resolution was guided by the Station ALOHA time-series deep trap collections, which were conducted over 12 day intervals during deployments from Mar 2014 through Nov 2016. We have added further discussion about the temporal change of b coefficients from power-law curves along with their correlations with environmental parameters in the revision. Please text lines 131-143.

The depth-specific change of SASVs showed two patterns, the decrease in the upper water and the increase in the deep water. One is in the euphotic zone and the other one is in mesopelagic and bathypelagic zone. The particle formation and export processes are different within euphotic zone vs. mesopelagic and bathypelagic zones. Please check the text lines 233-257 in the revision.

Ranges were provided in text lines 91-97, 100-102, 148-154, 190-197 in the revision.

L292-294: Why were individual time points from 4000m traps compared to monthly averages of water column ASV abundances and not the same time points? To account for sinking lag times? Were the sample dates not coordinated?

4000 m traps have a greater temporal resolution with individual samples being collected every 12 days (text line 278). Station ALOHA time-series water column samples on the other hand, were collected monthly (text line 296). These were the available data. The sample timing was dictated by practical and logistical considerations inherent in a complex field study such as this.

L297: This 500 m trap is from the Paragon cruise, right?

Correct. This is indicated in lines 300 and 315-317 in the revision.

L306-308: I read this sentence at least 4 times and I'm not totally sure I got it. Please reword.

Thanks for this suggestion. We do agree, as written this was difficult to interpret. In the revision, we have clarified this section as follows: “A total of 212 SASVs were identified as having a single depth maximum, while 47 SASVs had bimodal depth maxima. For these SASVs having bimodal peaks, 41 SASVs were retained for the following analyses only if: 1. Only one of the bimodal depth maxima exceeded the depth-averaged water column relative abundance; or 2. Both depth maxima were present in a single depth zone, as categorized in Figure 3.”

L313-317: These sentences seem non-sequitur. Why the sudden emphasis on intra- vs. extracellular DNA?

We have deleted these sentences in the revision.

Was the Paragon data combined with the other ALOHA data, or were these kept separate for different analyses? Please detail how they were used together or separately.

The Station ALOHA time-series 4000 m trap samples were compared with Station ALOHA time-series water column samples to obtain 4000 m trap SASVs. The PARAGON PIT trap samples were compared with PARAGON water column samples to get 500 m, 300 m, 250 m, 225 m, 200 m and 175 m trap SASVs. We clarified this in text lines 357-359 and 371-375.

REVIEWERS' COMMENTS

Reviewer #1 (Remarks to the Author):

I carefully read the revised manuscript. The paper has improved a great deal.

The paper makes a very impressive contribution to our understanding of the biological carbon pump in terms of planktonic microbial signatures, covering a beautiful time and depth scale for the ALOHA site.

The picture of the temporal and depth resolution of the microbial observations is the most important outcome of the study, I think, and not the potential use of SASVs as tracers of the carbon pump. We can perfectly follow the carbon pump by assessing POC attenuation and changes in nutrient and DIC concentrations.

I support acceptance of the manuscript.

Reviewer #2 (Remarks to the Author):

I have reviewed the Author Responses to both Reviewers, the revised manuscript, and all associated files. I feel the authors have done a very good job addressing Reviewer concerns and I have no further comments that need addressing. I found the process of reviewing the comments and the new manuscript highly educational, so I thank the authors for taking the time to so thoroughly examine all reviewer comments and make the changes that they did.